# Reinforcement Learning-Enabled Environmentally Friendly and Multi-functional Chrome-looking Plating

**Taigao Ma#**
Department of Physics
University of Michigan, Ann Arbor
taigao@umich.edu

**Anwesha Saha#**
Department of Applied Physics
University of Michigan, Ann Arbor
anweshas@umich.edu

**Haozhu Wang**
Department of EECS
University of Michigan, Ann Arbor
hzwang@umich.edu

**L. Jay Guo\***
Department of EECS
University of Michigan, Ann Arbor
guo@umich.edu

## Abstract

Although decorative chrome plating (DCP) is ubiquitous in metal finishings and coatings, the industrial process of chromium deposition is fraught with adverse health effects for the workers involved and causes environmental pollution. In this work, we seek to find an environmentally friendly replacement to DCP by mimicking the chrome color used for decoration. To discover a suitable replacement efficiently, we employ a reinforcement learning (RL) algorithm to perform an automatic inverse design in optical multilayer thin film structures. The RL algorithm successfully figures out two different structures with environmentally friendly materials while still showing a chrome color. One structure is further designed to have high transmission in the radio frequency regime, a property that general metals cannot have, which can broaden the decorative chrome applications to include microwave operating devices. We also experimentally fabricate these structures and validate their performance.

## 1 Introduction

Optical multilayer thin film structures are commonly used in many photonic applications, including solar cells[1], OLEDs[2], radiative cooling[3], etc. They have also been extensively used to produce long-lasting structural color coatings[4, 5], in contrast to traditional colors and coatings that are made from organic dyes or colored pigments. One such color coating is Decorative Chrome Plating (DCP), a metal finishing process widely used in the industry for coating various automobile parts, kitchen appliances, plumbing fixtures, etc., because of its distinct aesthetic and shiny reflecting appearance.

However, the chromium (Cr) layer deposition process in the industry for DCP is not a sustainable process and can cause adverse health impacts that can significantly outweigh its benefits, e.g., increase the risk of lung, nasal, and sinus cancer [6, 7, 8]. During fabrication, toxic emissions containing cadmium and cyanide released during the electroplating process can also lead to air pollution, which could impact the health of millions of people. In addition, DCP are often found on vehicle bodies,

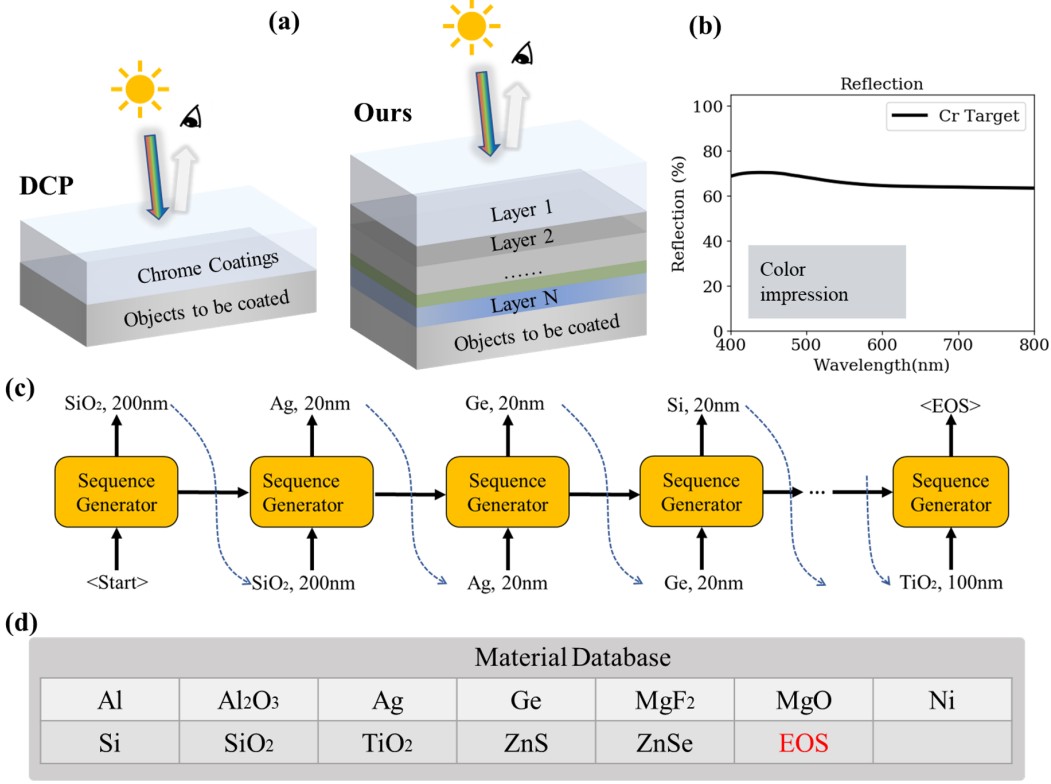

Figure 1: (a) Illustration of the Cr-replacement multilayer structures design. The left schematic shows a normal reflection diagram of a Cr-coated object, and the right schematic shows an N-layer thin film structure on a coated object which is used to mimic the reflection of the Cr-coated object while excluding Cr from the material selection. (b) The reflection spectrum and visual appearance of a 50 nm Cr layer coated. (c) Diagram of the sequential design process used by OML-PPO algorithms. (d) The user-defined material database for the RL algorithm to select materials.

especially emblems. Despite the attractive appearance, such shiny metal-based emblems block the radio frequency (RF) transmission needed for many vehicle sensors[9, 10]. For this, it would be desirable to eliminate the usage of chromium (metal) in the plating process while still preserving its attractive appearance.

In this work, we propose to remove the chromium element in DCP by mimicking the visual appearance of Cr-plated objects using multilayer thin film structures. We use the reinforcement learning algorithm, Optical MultiLayer-Proximal Policy Optimization (OML-PPO)[11], to finish the design and find two structures that visually look like Cr-plated objects, but without using chromium. One of the structures is designed to be transmissive in the radio frequencies (RF) while retaining the metallic appearance of chromium, which enables the extension of DCP to RF applications. This presents a way in which machine learning can be combined to solve scientific and engineering challenges and provide an environmentally sustainable approach to produce attractive metallic-looking coatings.

## 2 Methods

### 2.1 Problem Set

In order to mimic the visual appearance of Cr-coated objects, we need to design a multilayer structure that has a similar reflection spectrum as chromium's. We illustrate this idea in Fig. 1a and show the target reflection spectrum and visual appearance of a Cr-coated object in Fig. 1b.

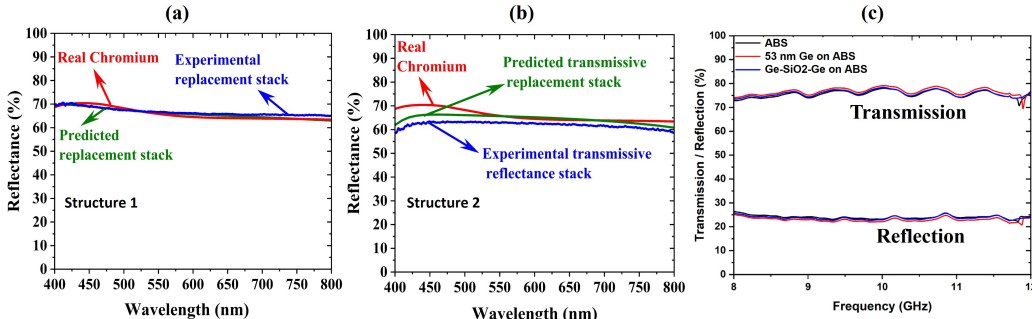

Figure 2: Comparison of the target reflection spectrum of Cr (red), the measured (blue) and simulated (green) reflection spectrum of two designed structures: S1 (a) and S2 (b). (c) The measured transmission and reflection spectrum of S2 in the RF region coated on the ABS substrate.

Consider a multilayer thin film structure as shown on the right side in Fig. 1a, we can express each layer as $s_l = [m_l, d_l]$, where $m_l, d_l$ correspond to the material and thickness at the $l_{th}$ layer. Then we can use the sequence $S = \{s_1, s_2, \ldots, s_N\} = \{[m_1, d_1], [m_2, d_2], \ldots, [m_N, d_N]\}$ to describe the overall thin film structure, where $N$ is the total number of layers. Therefore, designing such a structure is to find a specific sequence of material and thickness combination that gives the desired target optical response, which can be treated as a sequence generation problem.

## 2.2 Reinforcement Learning

Here, we use the RL algorithm OML-PPO[11] to finish the design process. The design step is illustrated in Fig. 1c. When designing the $l_{th}$ layer, the RL algorithm takes the designed layers from previous steps and predicts the selection of material $m_l$ and thickness $d_l$ sequentially. The thickness is selected from 50 different choices, ranging from 5 nm to 250 nm with discretization of 5 nm gap. The material $m_l$ is selected from a database that contains twelve different materials (see Fig. 1d). The design process will automatically stop when the designed structure reaches the maximum number of layers $N$ (here we set $N = 5$). We also have one faux material that is used as an indicator of the End-Of-Sequence (EOS). Once EOS is selected, the design process stops immediately irrespective of whether the designed structure has reached the maximum number of layers. This can be used to design structures with a variable number of layers (for example, 3, 4, 5 layers when $N = 5$).

During the design process, we use the Transfer Matrix Method (TMM)[12] to evaluate and simulate the reflection spectrum $R_d$ of the designed structure and aim to minimize the difference between the target spectrum $R_t$ and the designed spectrum $R_d$. Since the RL algorithm learns to solve problems by maximizing the reward, we define our reward $G_t$ to be 1 minus the spectrum difference and express the reward as:

$$G_t = 1 - \frac{1}{n} \sum_{i=1}^{n} (R_t(\lambda_i) - R_d(S, \lambda_i))^2 \tag{1}$$

where $\lambda_i$ is the wavelength which ranges from 400 nm to 800 nm with 5 nm increments. The reward will be calculated at the final step (meaning that the structure $S$ has been completely generated). After the design process, we select the best-discovered structure and use Particle Swarm Optimization (PSO) to further finetune the thickness at each layer, which can eliminate the influence of thickness discretization and improve the performance.

## 3 Results

### 3.1 Optical Performance

Using the RL algorithm described above, we find many structures as potential replacements for DCP and decide to experimentally fabricate two structures for their ease of fabrication. Several

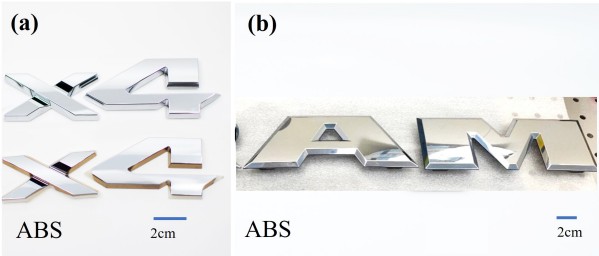

Figure 3: Visual comparison of different structures coated with commercial DCP (top in a), S1 (bottom in a), and S2 (b). The coating is on the ABS plastic.

other designed structures can be found in [13] with simulation results. We denote these two selected structures as 'Structure 1' (S1) and 'Structure 2' (S2). Going from the top layer to the bottom layer, S1 includes Ge (19 nm) / $TiO_2$ (17 nm) / $SiO_2$ (82 nm) / Ni (139 nm) and S2 includes Ge (21 nm) / $SiO_2$ (119 nm) / Ge (33 nm). Since there are no metals in S2, it can potentially allow high transmission in radio frequencies, providing a multi-functional property to DCP for RF applications.

We experimentally deposit these two structures using electron beam evaporation. In Fig. 2(a) and 2(b), we show the predicted (green) and measured (blue) reflection spectrum at normal incidence of S1 and S2, respectively. The spectrum is measured using a normal incidence spectrometer. These discrepancies between experimental and simulated results can be attributed to slight thickness variations during deposition. We can see that the measured reflection spectrum is very close to the target spectrum. In Fig. 3, we further compare the visual appearance of these two structures with the commercial DCP, demonstrating that our designed structures can provide the same decoration purpose as DCP.

### 3.2   Multifunctionality in the RF region

Commercial DCP-coated objects block RF signals because metals (chromium in DCP) are highly reflective in such frequencies. Thus, they are seldom used in RF-based devices, e.g., automobile radars. Semiconductor materials such as Ge have much smaller losses in the RF regime[14], so we design S2, another chrome-mimicking structure that has high transmission in the RF regime simultaneously, as the structure involves thin layers of Ge and $SiO_2$ only. To demonstrate this, we experimentally measure the transmission and reflection of S2 in the range of 8-12 GHz as shown in Fig. 2c. We test the RF transmission of (1) the substrate without any coating, (2) the substrate coated with a 53 nm layer of Ge (the combined thickness of the two Ge layers in S2), and (3) the substrate coated with S2. By analyzing the S-parameters. we find the transmission and reflection remain very close to that of a pristine substrate, suggesting that the designed S2 is a multi-functional coating that yields a similar appearance as DCP, while providing high transmission in the RF region, which is impossible using traditional metal-based chrome coatings.

## 4   Conclusion

Using reinforcement learning as a powerful inverse design algorithm, we find two multilayer thin film structures that can replace DCP by mimicking its visual appearance. Because these two structures do not contain any chromium, their fabrication will not cause health impacts or pollution issues, making them environmentally friendly to be used in real life. In addition, one structure exhibits high transmission in the RF region, a function that general Cr coatings cannot have. This feature can also expand the usage of DCP to RF devices. All of these features are empowered by the RL algorithms for inverse design, demonstrating how machine learning can be combined to solve practical scientific and engineering problems and help to contribute to sustainable and environmentally friendly industrial applications.

## 5 Acknowledgement

This work has been supported by the Michigan Translational Research & Commercialization (MTRAC) program and the National Science Foundation under grant No. PFI-2213684.

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
