# OpenReview forum: "Reinforcement Learning-Enabled Environmentally Friendly and Multi-functional Chrome-looking Plating"
_NeurIPS.cc/2023/Workshop/AI4Science — NeurIPS2023-AI4Science Oral_

### Official Review · Reviewer_9k4W · 2023-10-24
**Sequence generation design using a previously published model for multilayer thin film on a new application, decorative chrome plating**

**Rating:** 7
**Confidence:** 4

**Review:**

The authors train a deep sequence generation network to perform an inverse design in optical multilayer thin film structures for coating
with a shiny reflecting appearance. Two structures are designed.  The second structure further has high transmission in the radio frequency regime. The authors further experimentally validate the appearance of the two materials and measure the transmission and reflection of radio frequencies for the second material. Although the significance of the materials is unclear as stated below, I suggest accepting it, because experimental validation is not easy.

From the Introduction, the authors first justify the significance of this work as providing "an environmentally sustainable approach to produce attractive metallic-looking coatings", such that multilayer thin films can replace currently widely used Decorative Chrome Plating (DCP) with chromium (Cr). However, the resulting best films contain germanium (Ge). Considering the price of Ge is 100 times more expensive than Cr, it's hard to imagine one would use Ge and an intensive deposition method to replace the current coating process.

The authors also mention that "shiny metal-based emblems block the radio frequency (RF) transmission needed for many vehicle sensors.", but no references are provided. To what extent this is an issue?

---

### Official Review · Reviewer_9mBn · 2023-10-24

**Rating:** 8
**Confidence:** 3

**Review:**

The authors present a RL-PSO workflow to design a multi-layer coating recipe to replace Cr coating. The method is clearly articulated, and the design result is experimentally verified.
The only thing I feel unclear about is why the authors choose to design the layers sequentially, instead of encoding the recipe as a vector and then optimize all layers at the same time. The latter should better avoid falling into local minima during the PSO.

---

### Meta-Review · Area_Chair_MAdd · 2023-10-27

**Recommendation:** Accept (Oral)
**Confidence:** 4

**Metareview:**

This paper studies chromium deposition, an important topic for environment and sustainable development. Authors use an RL algorithm, to successfully develop two new structures. Further wet-lab experiments verify their validness. This is a theory-to-lab pipeline, and is very exciting to see how to expand this for wider range of tasks.